# The Determination of Assistance-as-Needed Support by an Ankle–Foot Orthosis for Patients with Foot Drop

**DOI:** 10.3390/ijerph20176687

**Published:** 2023-08-30

**Authors:** David Scherb, Patrick Steck, Iris Wechsler, Sandro Wartzack, Jörg Miehling

**Affiliations:** Engineering Design, Friedrich-Alexander-Universität Erlangen-Nürnberg, 91058 Erlangen, Germanymiehling@mfk.fau.de (J.M.)

**Keywords:** ankle–foot orthosis, digital human models, gait assistance, musculoskeletal simulation, foot drop, muscle weakness, design for medical devices

## Abstract

Patients who suffer from foot drop have impaired gait pattern functions and a higher risk of stumbling and falling. Therefore, they are usually treated with an assistive device, a so-called ankle–foot orthosis. The support of the orthosis should be in accordance with the motor requirements of the patient and should only be provided when needed, which is referred to as assistance-as-needed. Thus, in this publication, an approach is presented to determine the assistance-as-needed support using musculoskeletal human models. Based on motion capture recordings of multiple subjects performing gaits at different speeds, a parameter study varying the optimal force of a reserve actuator representing the ankle–foot orthosis added in the musculoskeletal simulation is conducted. The results show the dependency of the simulation results on the selected optimal force of the reserve actuator but with a possible identification of the assistance-as-needed support required from the ankle–foot orthosis. The required increase in support due to the increasing severity of foot drop is especially demonstrated with the approach. With this approach, information for the required support of individual subjects can be gathered, which can further be used to derive the design of an ankle–foot orthosis that optimally assists the subjects.

## 1. Introduction

Foot drop is a physiological dysfunction affecting the lower leg muscles surrounding the ankle joint [1,2], possibly being caused by multiple diseases like stroke [3], multiple sclerosis [4], cerebral palsy [5] or Charcot–Marie–Tooth disease [6]. The outcome is an influence on the lower leg muscle functions, which can either result in a complete loss of function (paralysis) or in a partial loss of function (paresis) of the individual muscle [7]. Mainly, the dorsiflexors, comprising the muscles at the shin responsible for lifting the foot (dorsiflexion), are paralyzed. The counterpart of the dorsiflexors is the plantarflexors (muscles at the calf) which are accordingly responsible for lowering the foot (plantarflexion). These muscles are affected to an individual extent, ranging from no implication over paretic influence to paralysis of the muscles. In any case, these muscle deficiencies cause the patient to have a foot drop expressed in the inability to lift the toes and having them just hanging towards the ground [2,8]. The patients’ issues with foot drop are especially manifested during gait. Patients show an altered gait pattern called “steppage gait” [9] with higher hip and knee flexion or a rotational compensatory movement of the whole leg [6,10] to obviate the instability in the ankle and knee joint, which leads to a higher risk of stumbling and falling [11,12]. Furthermore, patients suffering from foot drop show a slower gait speed [10,13] and a slap of their foot on the floor instead of a smooth and controlled lowering movement [14].

There are several options to treat foot drop depending on the patient’s condition, such as surgery [15] or functional electric stimulation [8,16]. The most common one so far is still the treatment with an ankle–foot orthosis (AFO). An AFO is a device externally applied to the foot and lower leg that supports humans in daily life by force transmission and providing stability in the ankle joint [17,18,19]. The main aim of an AFO is to prevent foot drop and preserve a human’s gait such that it can be referred to as a regular one. Studies prove that patients with an AFO have increased stability in the ankle joint [20], higher gait speed [20,21], improved dynamic balance control [21], increased levels of self-esteem [22], rectified energy consumption during walking [23] and, overall, a better gait [20,21,23]. In general, Shorter et al. [24] define four requirements an AFO has to fulfill in order to realize an optimized treatment for the patient, which are directly related to the occurring events during the gait cycle (see Figure 1):Support of dorsiflexion at heel strike to prevent foot slap;Free plantarflexion during foot flat and midstance phase to provide stability for the knee and ankle;Support of plantarflexion to realize push-off before toe-off;Support of dorsiflexion in the swing phase to prevent toe drop.

These requirements should ideally be in accordance with the user’s assistance needs depending on the muscle power loss to realize an optimized treatment for the patient.

However, the synchronization of required and provided support is hard to realize in a purely empiric design process due to the lack of recordable information in user tests [25,26], ethical restrictions limiting user tests [27] and the cost-and-time-consuming design revision process [28]. Therefore, for the design of orthoses and similar devices (referred to as “wearable assistive devices”—devices that support or protect the human body by force transmission)—digital human models, mostly musculoskeletal human models (MHMs), are often used [29]. These models allow the representation of the effects on the biomechanics of the patient executed by the wearable assistive devices, like the possible decrease in required metabolic energy during lifting heavy objects by using a knee support [30] or the effects on the muscle activation by an upper body power assist suit [31]. Studying the effects of orthoses on the human body with MHMs provides valuable insight into their functionality and can be used to optimize the design of the devices. So, the effects of orthosis properties, like stiffness, damping, etc., on the required lower leg muscle force [32,33,34], as well as the reduction in plantarflexor muscle load and metabolic energy consumption during hopping with the aid of ankle assistance [35], the effects of different ankle exoskeleton strategies for plantarflexor muscle energetics [36] or the predictions of walking cost reductions with ankle assistance [37], have already been investigated.

Furthermore, musculoskeletal simulation can be used to predict the ideal support of an AFO to prevent foot drop. Depending on the remaining muscle force in the lower limb, the orthosis should only adopt a specific part of work and instead primarily utilize the remaining muscle power. Thereby, a further decrease in muscle condition can be impeded. This kind of assistance is referred to as assistance-as-needed (AAN). Accordingly, Carmichael et al. [38] defined the required assistance of arm support devices for grasping tasks based on the occurring torques in healthy and weakened models. Also, Afschrift et al. [39] used the AAN approach to identify the capability gaps during different activities of daily living (stand up/sit down, walking stairs, etc.) at varying muscle weaknesses in the whole lower limb. In their study, the aim was to identify the timing and magnitude when assistance is required (depending on the present muscle weakness and the investigated activity). However, an even muscle weakening in the whole lower limb was executed during their investigations. Foot drop, as previously described, mainly affects the lower leg muscles, and the muscle weakness is present in different severity combinations between both affected muscle groups (dorsi- and plantarflexors).

Thus, the aim of this study is to show an approach for defining the AAN support ideally provided by an ankle–foot orthosis for patients suffering from foot drop and, ultimately, enabling the best possible treatment for these patients. In order to identify this assistance, an approach is required to calculate the optimal support for patients. Therefore, a variety of subjects, gait speeds and pathological situations (different severities of foot drop) are studied to acquire a potential broad representative group of patients as a data basis for the design of a passive AFO which shall be capable of an optimized treatment for foot drop patients [40,41]. Thus, the research question is, how can an approach be realized for determining the optimal AAN support by an AFO for different patients, gait speeds and foot drop severities?

## 2. Materials and Methods

In this section, the executed actions for setting up the approach to identify the optimal AAN support by an AFO for a variety of requirements are presented. First, MHMs of the participants that should be able to replicate the subjects’ principle biomechanical performance are created. Second, the MHMs are manipulated to represent foot drop patients and equipped with a virtual assistance in the simulation to reproduce the required support from the AFO. The optimal support by the AFO is calculated via a parameter study.

### 2.1. Biomechanical Analysis of Healthy Subjects

For identifying the occurring healthy muscle loads, muscle activation patterns and torques in the lower leg during gait, gait data of different healthy subjects are captured in a motion capture laboratory. The data are used to create MHMs of the subjects and further to conduct the biomechanical analysis.

#### 2.1.1. Experimental Data

Fifteen healthy subjects (age: 30 ± 12 years, height: 1.76 ± 0.1 m, weight: 72 ± 11 kg, BMI: 23 ± 1.8 kg/m^2^, 7 females, 8 males) participated in this study and provided written informed consent in accordance with the ethics commission of the FAU Erlangen-Nürnberg. The subjects were recorded in a gait laboratory with an optical motion capture system with 10 cameras (Qualisys Motion Capture Systems, Gothenburg, Sweden, SW) sampled at 100 Hz. The subjects were recorded in a calibrated volume of 8 m × 3 m × 2.5 m with an average residual of less than 1 mm for each camera. The markers were attached to bony landmarks and required body parts according to the Plug-in-Gait model (Vicon Motion Systems, Oxford, UK), which can be seen in Figure 2. Each subject performed three gaits at varying gait speeds (slow, medium/normal, fast) and selected the individual speed they felt comfortable with themselves. The three gait speeds were recorded at all three times. The resulting average speeds for all subjects are the following:Slow: 0.97 ± 0.17 m/s;Medium: 1.25 ± 0.23 m/s;Fast: 1.71 ± 0.21 m/s.

The ground reaction forces were measured using two force plates (Bertec GmbH, Columbus, OH, USA) sampled at 100 Hz. The muscle activity of the soleus, gastrocnemius medials and tibialis anterior of right body side were measured using wireless surface electromyography (EMG, Noraxon Desktop DTS, Scottsdale, USA) sampled at 1500 Hz. As for electrodes, single N-electrodes were used (Ambu AG, Switzerland, Dietikon, CH). For placement of the electrodes, SENIAM recommendation was used [42]. The EMG data were band-pass filtered, rectified and low-pass filtered with a Butterworth filter and a cutoff frequency of 6 Hz.

#### 2.1.2. Musculoskeletal Modelling and Simulation

Musculoskeletal simulation was carried out with OpenSim 4.2 (OpenSim, Stanford, CA, USA) [43]. A generic full-body MHM was used for investigation [44] and was, in the first step, scaled to the anthropometry and weight of the subjects. In the second step, the force-generating capacity of each subject model was adjusted, i.e., the properties of the muscles were adjusted. The optimal fiber length and tendon slack length of each muscle were adjusted according to the anthropometric scaling of the model as part of the regular scaling process in OpenSim. For adjusting the maximum isometric force of each muscle, another process was applied. Handsfield et al. [45] discovered a relation between the total lower limb muscle volume (V_mus_^tot^) and the body height (h) and mass (m) of a healthy, young person:V_mus_^tot^ = 47∗m∗h + 1285 (1)

Additionally, the average fraction of every individual muscle volume (φ_mus_) from this total lower limb muscle volume was calculated for all subjects in their study. Therefore, the single muscle volumes of the subjects in our study can be computed based on the subject’s body height and mass. Then, the maximum isometric force of a muscle (F_mus_^0^) is computed with the following equation, also used by Rajagopal et al. [46] in their model:(2)$${F_{\mathrm{mus}}}^{0}=\sigma_{\mathrm{mus}}\frac{\phi_{\mathrm{mus}}{v_{\mathrm{mus}}}^{\mathrm{tot}}}{{l_{\mathrm{mus}}}^{0}}$$

The optimal fiber length (l_mus_^0^) of each muscle results from the anthropometric scaling, as previously mentioned, and the specific tension (σ_mus_) is assumed to be 60 N/mm^2^, as chosen by Rajagopal et al. [46]. The result is an adjusted muscle force-generating capacity for every subject based on body height and mass. However, in the muscle scaling process, only young, healthy subjects have been considered so far. Accordingly, no regression in muscle force over peoples’ lifespan is incorporated, which normally reaches its maximum around 30 years of age and decreases with increasing age [47,48]. Thus, the strength decline is incorporated by weakening the maximum isometric force of each muscle, according to the study of Alcazar et al. [49], depending on the age group of the subject.

The finalized scaled models (Figure 2) of the subjects reproduced the measured gaits by means of the inverse kinematics algorithm that minimizes squared distances between measured markers and virtual markers. From each gait (for each patient and each gait speed), one gait cycle starting from the right heel strike on the first force plate to the next right heel strike was extracted and considered for further investigation. Due to the missing ground reaction force on the left foot at the first time frames (because of the non-availability of a third force plate), the ground reaction force was estimated in OpenSim with point and torque actuators (in *x*, *y* and *z* direction) applied to left calcaneus body in the subject models. Then, to minimize the dynamic inconsistencies between motion data and external force data, the residual reduction algorithm [50] was conducted. Based on the smoothed motion data and external force data, the occurring joint torques were determined with inverse dynamics. The torques were then normalized to the mass of the subject. The required muscle forces and activations of the subject models for performing the motions were calculated using static optimization (SO) in OpenSim. The objective function of the optimization solves the required muscle forces for an existing joint torque (τ). Due to the normally occurring overdetermination, i.e., multiple muscles (mus) are responsible for one joint torque (referred to as muscle redundancy), an equation has to be solved at every frame (j) to determine the occurring muscle activation (a_mus_) depending on the lever arm (r) and maximum isometric force (F_mus_^0^) of the muscle:(3)$$\sum_{\mathrm{mus}=1}^{n}\left(a_{\mathrm{mus}},{F_{\mathrm{mus}}}^{0}\right)r_{\mathrm{mus},j}=\tau_{j}$$

The objective function of static optimization finally solves the muscle redundancy problem by minimizing the sum of muscle activation squared, resulting in the performance indicator (J):(4)$$J=\sum_{\mathrm{mus}=1}^{n}{\left(a_{\mathrm{mus}}\right)}^{2}$$

Accordingly, a distribution of muscle activation and force, which depends mainly on maximum isometric force of the muscle, is realized, as it also occurs in reality. The resulting simulated muscle activation dynamics from soleus, gastrocnemius medialis and tibialis anterior are then compared to measured EMG data to evaluate a correctly executed and valid simulation.

### 2.2. Determination of Required AFO Assistance

For defining the required AFO assistance, a workflow shown in Figure 3, which is described in detail in this section, is applied.

#### 2.2.1. Creation of Foot Drop Patient Models

In the first step, MHMs representing patients with foot drop have to be created. As a result of the occurring paralysis or paresis in the lower leg muscles, the force each muscle can apply is affected. More specifically, either the muscle is unable to create force (paralysis) or the maximum force of the muscle is weakened (paresis). As a consequence of foot drop, the dorsiflexors are paralyzed. Accordingly, the maximum isometric force of the muscles classified to the group of dorsiflexors is set to 0 N on the right side of the subjects’ models. For visualization purposes, the dorsiflexor muscles are colored grey in Figure 4. The effect on the plantarflexors is different in individual cases, ranging from no implications to paralysis of the muscles. Thus, the maximum isometric force is decreased in four steps from 100% maximum isometric force (named PF100) to 0% maximum isometric force (PF0). The plantarflexors are colored blue in Figure 4. Concluding, condition PF75 indicates a foot drop patient model of a subject with 0 N maximum isometric force of the dorsiflexors and 75% remaining force in the plantarflexors (equal to 25% weakening). Finally, five foot drop MHMs of each recorded subject (from now on referred to as patients) with different severity of foot drop on the right side are created.

#### 2.2.2. Simulation of AFO Assistance

As the next step, the assistance provided by the AFO for the foot drop patient models has to be implemented in the simulation. Our hypothesis here is that an optimal support of the AFO for the patient results in a gait equal to the gait of a healthy subject (stride length, movement pattern, metabolic cost, etc.). Accordingly, the muscle activation of the lower leg muscles in healthy conditions also has to be equal to the muscle activation of the lower leg muscles in weakened conditions with AFO assistance. Thus, the SO is also conducted with the foot drop patient MHMs, in combination with the gait and ground reaction force data for all three speeds recorded from the healthy subjects. In addition, the support of the AFO during gait is implemented via a so-called “reserve actuator”, which is an additional virtual force in OpenSim acting ideally along the axis of a generalized coordinate (in this case, the ankle joint). During the definition of the reserve actuator, an “optimal force” has to be assigned, which can be considered similar to the maximum isometric force assigned in the muscle model definition. Thus, the reserve actuator is also respected in the goal function of the SO (Equation (4)), and its contribution depends largely on the chosen optimal force. For comparison, the effects on the plantarflexor muscles in the PF50 condition of one exemplary subject are shown for a reserve actuator optimal force of 1 Nm (Figure 5a) and 10,000 Nm (Figure 5b). Here, it can be clearly seen that the calculation of the plantarflexor muscle activation depends heavily on the chosen optimal force of the ankle reserve actuator. Therefore, a suiting value for the ankle reserve actuator optimal force has to be identified for every patient and condition, resulting optimally in a muscle activation equal to the muscle activation in the healthy condition.

#### 2.2.3. Parameter Study for Identifying Optimal Orthosis Support

In order to identify the optimal AFO support for every patient, gait speed and weakened condition, a parameter study varying the ankle reserve actuator optimal force is conducted. The optimal force is varied in steps of 20 Nm from 20 Nm to 200 Nm, which is chosen because 200 Nm is the maximal occurring ankle torque of all subjects and gait speeds. Since the target is the plantarflexor muscle activation of the subject in the healthy condition at the investigated gait speed, the calculated plantarflexor muscle activation of the varied ankle reserve actuator simulation is compared to it. Thus, at first, the average deviation of every plantarflexor muscle (x_mus_) is computed over the gait cycle, i.e., over all time steps (t). This is realized by calculating the difference in muscle activation in the weakened situation with reserve actuator support at the specific parameter of optimal force (a_mus,parVar_) and the healthy muscle activation (a_mus,healthy_):(5)$$x_{\mathrm{mus}}=\frac{1}{t}\sum \left(a_{\mathrm{mus},\mathrm{parVar}}\;-\;a_{\mathrm{mus},\mathrm{healthy}}\right)$$

Then, the average deviation of the specific parameter of optimal force for all plantarflexor muscles (x_parVar_) is computed over all plantarflexor muscles (mus):(6)$$\;x_{\mathrm{parVar}}=\frac{1}{\mathrm{mus}}\sum \left(w\;*\;x_{\mathrm{mus}}\right)$$

w = 10: soleus, gastrocnemius medialis, gastrocnemius lateralis;

w = 1: all other plantarflexors.

In addition, a weight factor (w) is added to consider the higher priority for matching the muscle activation curve of the main contributors of the plantarflexors (soleus, gastrocnemius medialis, gastrocnemius lateralis).

### 2.3. Analysis

The analysis of the conducted simulated results will be two two-fold. First of all, the results obtained from the participants, or, more precisely, from the MHMs derived directly from them, will be evaluated to ensure a correct representation of the biomechanical behaviour of the participants. On the one hand, the occurring ankle joint torque is investigated. Numerous studies have already shown the characteristic curve during gait with the main torque arising in the stance phase of gait with a maximum peak of around 1.5 times the torque normalized to the patient mass [51,52]. On the other hand, the simulated muscle activations from the MHMs are compared with the measured muscle activations of the participants via EMG. Comparing the muscle activations, it first has to be stated that the EMG-measured muscle activations were normalized to the respective maximal occurring activation in the simulated activation over the gait cycle due to the missing MVC measurement. Thus, the general curves of the muscles and mainly the activation timings during the gait cycle are compared, but not the exact curves and the magnitude of activation, as was also previously conducted in other studies [46,53]. Secondly, the simulated muscle activations with every optimal force for the reserve actuator from the parameter study will be analyzed. For each combination of patient, gait speed, weakened condition and optimal force value of ankle reserve actuator, the average deviation over all plantarflexor muscles from the healthy condition is calculated with Equation 6. Then, the optimal force parameter of the ankle reserve actuator with the lowest average deviation over all plantarflexor muscles has the best agreement with the healthy muscle activation for a specific patient, gait speed and weakened condition. Finally, the optimal assistance that has to be provided by the AFO is represented by the curve of the used ankle reserve actuator over the gait cycle.

## 3. Results

In this section, the applied methods and the obtained results are described. First, the correctness of the created MHMs from the subjects’ recorded gait data was investigated by evaluating the resulting ankle joint torques and comparing the occurring simulated muscle activations with the measured muscle activations via EMG of the subjects. Then, the optimal support by the AFO was calculated by computing the lowest percentage muscular activation deviation from the healthy condition over all parameter variations, i.e., muscle activations of parameter variation closest to the healthy condition. This was carried out for each patient, gait speed and foot drop severity.

### 3.1. Biomechanical Analysis of Healthy Gait

The resulting ankle joint torques for the three gait speeds of six different patients selected for representation are shown in Figure 6. The curves all show the same characteristics. After a small positive torque (dorsiflexion) at the beginning of the gait cycle, the torque is inverted (plantarflexion) and increases to its maximum at around 1.5 times the torque normalized to the patient mass. After reaching its maximum, the torque decreases pretty rapidly, and after a probable peak for dorsiflexion, a small dorsiflexion torque remains present until the end of the gait cycle. Furthermore, an increase in the occurring plantarflexion torque with increasing gait speed can be observed for all patients.

Observing the simulated lower leg muscle activation via SO over the gait cycles (Figure 7—orange line), it can be seen that the dorsiflexors (represented by the tibialis anterior) are mainly activated at the beginning of the gait cycle, i.e., the stance phase. Then, over the stance phase of gait, the plantarflexors (represented by gastrocnemius medialis and soleus) become primarily activated until the end of the stance phase. In the swing phase of gait, the dorsiflexors are again mainly activated. Considering the activation height of the lower leg muscles, it can be seen that the dorsiflexors are only activated in a lower range of up to 20% (slow gait speed) and 40% (fast gait speed), whereas the plantarflexors are more activated up to the peak of about 50% (slow gait speed) and 80% (fast gait speed), respectively. The increase in muscle activation from slow gait speed to fast gait speed can be observed for all three muscles. Comparing the muscle activations of the soleus, gastrocnemius medials and tibialis anterior to the measured EMG signals, the major features of the measured EMG signals occur in the simulated activations, especially the measured peaks of the muscles. So, the soleus and gastrocnemius medialis (the plantarflexors) at the end of the stance phase and the tibialis anterior (dorsiflexor) at the beginning of the stance phase and at the swing phase are equally represented in the simulation. However, there are also some notable differences between the measured EMG and the simulated muscle activations. The measured EMG data show pretty high activations in the swing phase for the plantarflexors, especially for the soleus, whereas the simulation shows almost no activation in this phase. Furthermore, the tibialis anterior has some small activations during the stance phase for all patients with different gait speeds, whereas the simulations also show no activation. Nevertheless, due to the representation of the main features of measured EMG signals, it can be assumed that the musculoskeletal modeling of the patients and the related conducted simulations have a good agreement with the real occurrences and, therefore, represent the real subjects in a reasonable way.

### 3.2. Parameter Study for AFO Assistance

The results of the performed parameter study with the approach of determination of the optimal assistance by an AFO are exemplarily shown for one subject (P11). The determination of the optimal assistance for the remaining subjects with the weakened conditions and the different gait speeds was calculated equally. These results can be found in the Appendix A (Appendix A).

Varying the optimal force value of the ankle reserve actuators in the parameter study causes different curves for the muscle activations in the gait cycle. For example, the resulting range of the soleus muscle activation by all incorporated different parameters (red) for the slow gait speed of subject P11 is shown for all created foot drop MHMs in Figure 8 compared to the healthy soleus activation (blue). The resulting range of soleus activation becomes wider with the increasing weakening of the plantarflexor muscles, i.e., having the widest range for PF25. Furthermore, it can be seen that the simulations with any varied optimal force of the ankle reserve actuator for PF100 affect a decrease in the required soleus muscle activation. For PF0, the addition of all reserve actuator optimal forces to the simulation results in no change for the soleus activation, which is all 0 due to the complete paralysis of the lower leg muscles.

The resulting plantarflexor muscle activations are further used with Equations (5) and (6) to calculate the average deviation over all plantarflexors muscles compared to the healthy condition for each parameter variation. The resulting average percentage deviations for each foot drop severity of P11 are shown in Table 1. Each row represents the average percentage deviation of one created foot drop MHM for all varied optimal forces of the ankle reserve actuator. The smallest deviation for every foot drop severity is marked green, whereas the worst one is marked red. Due to the non-present influence on the muscle activations for PF0, the average deviations have (nearly) no difference. For the other foot drop conditions, it can be seen that with the increasing weakening of the plantarflexor muscles, a higher optimal force for the ankle reserve actuator has to be chosen to result in the smallest average percentage deviation.

The lowest percentage value for each foot drop condition depicts the assistance resulting in plantarflexor activations closest to the healthy condition. Thus, the utilized ankle reserve actuator torque in these simulations is equal to the optimal support torque that has to be provided by the AFO. This support (red) always adopts a fraction of the operating ankle torque (blue) during the gait cycle, as illustrated in Figure 9. The amount of provided assistive torque thereby depends on the foot drop severity of the patient.

When plotting the support torque over the occurring ankle angle values during the gait cycle, a curve representing the required power from the reserve actuator in the different phases of the gait cycle results (Figure 10). The curves start with a small dorsiflexion torque at the first heel strike (HS1), followed by a little increase of dorsiflexion torque and then a consistent increase of plantarflexion torque until the peak at the heel-off (HO). After the HO, the required plantarflexion torque decreases again until the turning point of assistance direction at the toe-off (TO). Finally, from the TO to the end of the gait cycle (second heel strike—HS2), only a small dorsiflexion torque is provided by the reserve actuator. The provision of dorsiflexion torque in the small time frame after the HS1 and from TO to HS2 is similar for all foot drop patients. The main difference occurs in the provided plantarflexion torque, ranging from the highest peak at PF0 until no provided plantarflexion torque at PF100.

The results of parameter variation for the soleus muscle activation, average percentage muscle deviation, optimal support torque and angle–torque curves for the medium and fast gait speed of P11 show high similarity to the previously mentioned results of the slow gait speed. Therefore, only the optimal AFO support torque compared to the ankle joint torque (Figure 11 and Figure 12) and the angle–torque curves (Figure 13 and Figure 14) are shown for medium and fast gait.

## 4. Discussion

The aim of this contribution was to show an approach for determining the required optimal support by an AFO for different patients, gait speeds and plantarflexor weaknesses at foot drop. By adding a reserve actuator representing the AFO assistance with different optimal forces in the musculoskeletal simulation, the load on the foot-drop-affected muscles can be adjusted. The resulting muscle activation can accordingly have a wide range (Figure 8), with one parameter showing the best agreement with the target activations (in our case, the healthy muscle activation) for each foot drop severity. The dorsiflexion torque during gait is ideally completely taken over by the AFO due to the occurring paralysis of the dorsiflexor muscles (Figure 9 and Figure 10). An increased weakening of the plantarflexor muscles results in an increased required plantarflexor assistance from the AFO, ranging from almost no required assistance (PF100) due to the present full strength of the muscles to full assistance (PF0) due to the complete power loss/paralysis of the muscles. Furthermore, it can be seen that the required assistance at the highest torque during gait (at around 50% of the gait cycle) nearly equals the assumed weakening of the plantarflexors, e.g., at PF50, the orthosis provides about 50% of the present maximal plantarflexion torque, which matches the expectations one might have prior to the simulations. Moreover, by analyzing Figure 9 and Figure 10, two points for switching the assistance can be identified. First, the assistance of dorsiflexion is required to support the controlled lowering of the foot. Then, the assistance has to be switched to plantarflexion with an increase to the maximum occurring torque at the HO. After the HO, the required assistance decreases pretty rapidly until the assistance direction is again switched around the TO to dorsiflexion. In the end, the dorsiflexion is assisted in preventing the down-hanging of the foot and, thereby, a possible stumbling and falling of the patient. This provided assistance is in good agreement with the requirements established by Shorter et al. Additionally, the angle–torque curves (Figure 9 and Figure 11) also provide information about the required energy that has to be provided by the AFO represented as the integral under the curve. The main energy has to be provided after the HS1 and the TO, with the maximum at the HO. It is nicely shown that the energy that has to be saved (and later has to be provided) by the AFO decreases with less weakening of the lower leg muscles, i.e., less energy is required for PF100 than for PF0. Furthermore, the angle–torque curves can be used to identify material parameters and material compositions that store and transmit the force over the AFO to the patient. These curves have previously been used to determine the torsional stiffness of springs attached to wearable assistive devices [54]. Thus, this evaluation can aid the realization of the final AFO design.

In addition, a strength scaling method for the lower limb muscles has been applied that considers the subject’s strength depending on the age of the subject. The resulting maximal isometric forces of the muscles are in a similar range to the calculated forces from Rajagopal et al. [46] that were used in their model. The resulting ankle joint torques (Figure 6) and muscle activations also show a reasonable activation curve over the gait cycle, especially when being compared to the measured EMG data (Figure 7) and additionally in comparison with previously conducted analyses [51,55,56]. However, there was no elderly person in our study (age > 65 years) with an accordingly strong decrease in strength, which could possibly result in simulated muscle activations of 100%. But it is also possible that these subjects show an altered gait pattern, i.e., actively avoiding the high muscle load during “normal/healthy” gait, which has to be investigated in a future study. Moreover, the identified strength difference between males and females in the same age range in other studies [47,49] is only considered by the height and weight of the subjects. Accordingly, the average difference in maximal ankle torque between males and females of subjects classified in the same age group is only about 15% compared to 30% in another study. The reason could be the missing extra scaling of the muscle’s optimal fiber length (only adjusted due to the anthropometric scaling), which directly influences the resulting maximal isometric force. Smaller subjects have smaller optimal fiber lengths due to anthropometric scaling and, accordingly, compensate for the smaller muscle volume. Thus, the strength scaling has to be extended to the scaling of the optimal fiber length of the muscles. Generally, the results of the muscle activations are difficult to validate because no EMG with maximal voluntary contraction, resulting in the relative muscle activations compared to the maximal force in reality, was conducted. For validation and enhancement purposes of the strength scaling method, the validation with EMG and MVC data has to be conducted in the future, but this was not the primary aim of this study.

Further limitations of this study are that the assistance of the AFO is assumed to be acting ideally in the ankle axis due to the incorporated reserve actuator. Small relative motions between the AFO and the human lower limb will occur in reality, causing a varying application point for the force provided by the AFO [57]. The mass of the AFO is also not considered in the simulation, having an impact on the resulting muscle activations. Additionally, the calf consists of biological soft tissue on which the force is applied. This soft tissue causes energy absorption and accordingly influences the desired effect of the assistance [29]. Thus, all these influences have to be considered when realizing an AFO based on these results. However, the simulation already provides valuable insights into the human body that are elementary for the design of the AFO. Another limitation is our hypothesis that a gait of foot drop patients equal to a “healthy” gait will result in an ideal support of the AFO. The methodologically correct approach would be to predict the new gait resulting from the AFO assistance. This is, however, very time-intensive and computationally intensive to realize. Moreover, it was the intention to gain knowledge about the magnitude and course of required assistance by the patients to enable the derivation of design proposals for an AFO, which can be realized with the shown approach. The testing of the designed product is inevitable anyway. The final limitation is the specification of defined foot drop situations of the patients, resulting in defined weakened muscle situations. In reality, the exact severity of foot drop, i.e., the weakened muscle situation, is very hard or even impossible to identify. It was conducted to receive the range for the required torque provision, which influenced the realization of the AFO design later. For designing a patient-specific AFO, a special testing procedure will have to be defined in future.

## 5. Conclusions

In conclusion, this publication demonstrates an approach for determining the required, ideal support by an AFO during the gait cycle for a variety of patients, gait speeds and foot drop severities. With the help of musculoskeletal simulation, the assistance provided by the AFO affecting the resulting muscle activations in the lower leg muscles is investigated via a parameter study of the possible AFO strength. By identifying the assistance with the smallest deviation from the healthy muscle activation, the ideal support, referred to as AAN support, can be established. Based on these results, concepts of an AFO that are capable of realizing the assistance of both rotational directions (dorsi- and plantarflexion) in the ankle joint and accordingly provide the best possible treatment for foot drop patients can be further designed and also be tested for validation purposes.

## Figures and Tables

**Figure 1 ijerph-20-06687-f001:**
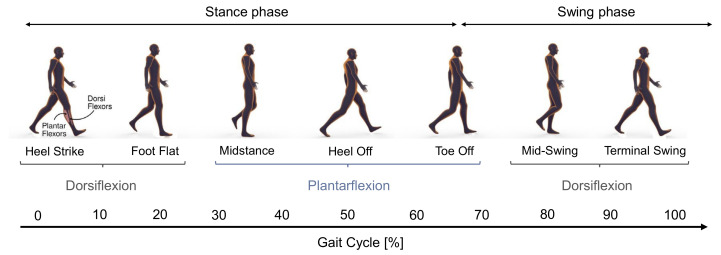
Specific phases, events and dorsi- and plantarflexion of right foot during gait cycle.

**Figure 2 ijerph-20-06687-f002:**
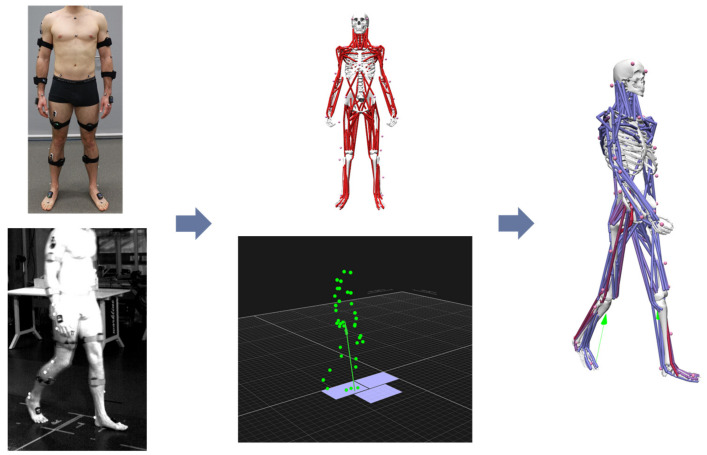
Experimental setup. Left images: Subject preparation with attached marker and EMG sensors (**top**) and recorded video of gait (**bottom**). Middle images: Scaled musculoskeletal model after marker location (**top**) and visualization of recorded gait data in Qualisys GUI (**bottom**). Right image: Resulting musculoskeletal simulation of gait with ground reaction force data (green) and resulting muscle load—blue: no/low load; red: high load.

**Figure 3 ijerph-20-06687-f003:**
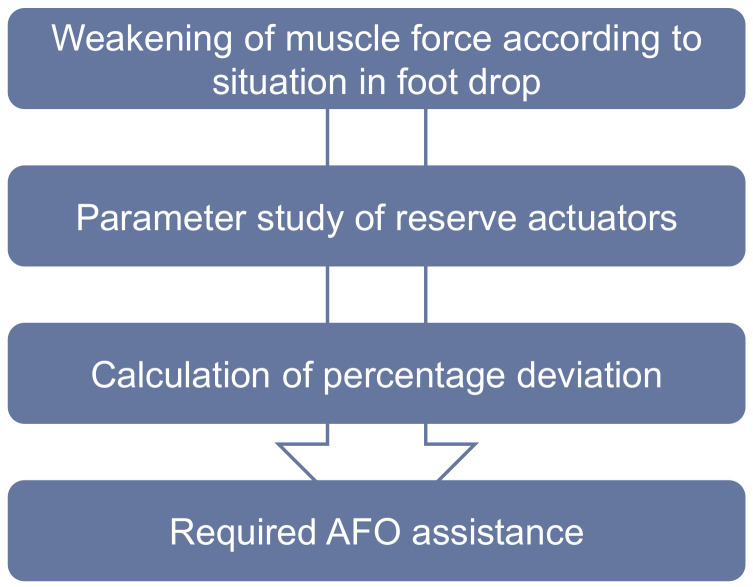
Workflow for determination of required AFO assistance.

**Figure 4 ijerph-20-06687-f004:**
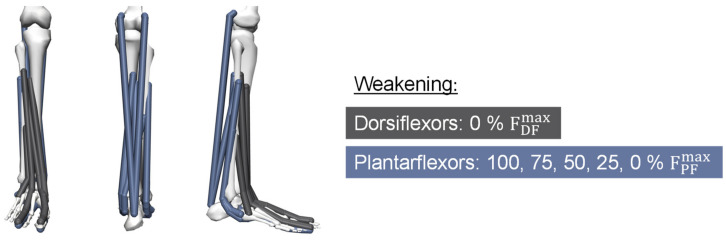
Weakening of dorsi- and plantarflexors in musculoskeletal human models.

**Figure 5 ijerph-20-06687-f005:**
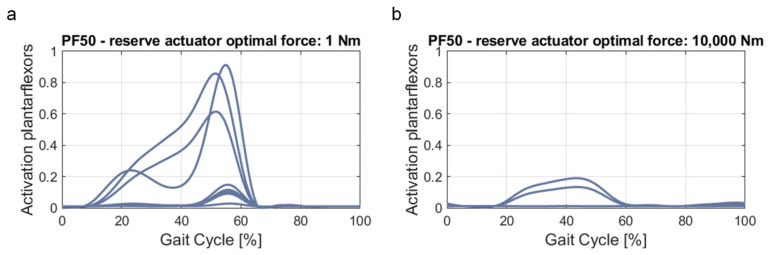
Resulting plantarflexors activation depending on the selected optimal force of the ankle reserve actuator: (**a**) plantarflexors activation at ankle reserve actuator optimal force of 1 Nm; (**b**) plantarflexors activation at ankle reserve actuator optimal force of 10,000 Nm.

**Figure 6 ijerph-20-06687-f006:**
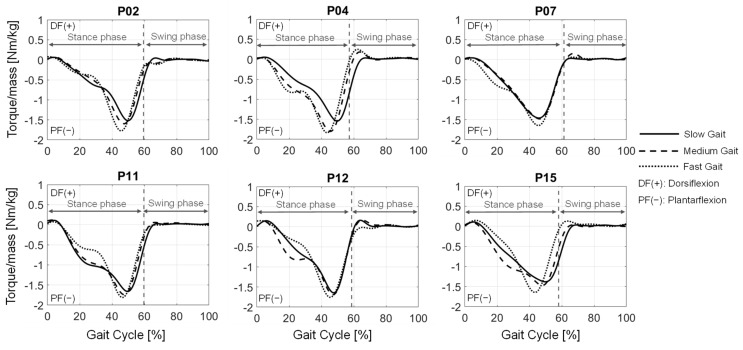
Ankle joint torques for different patients at the three gait speeds over the gait cycle.

**Figure 7 ijerph-20-06687-f007:**
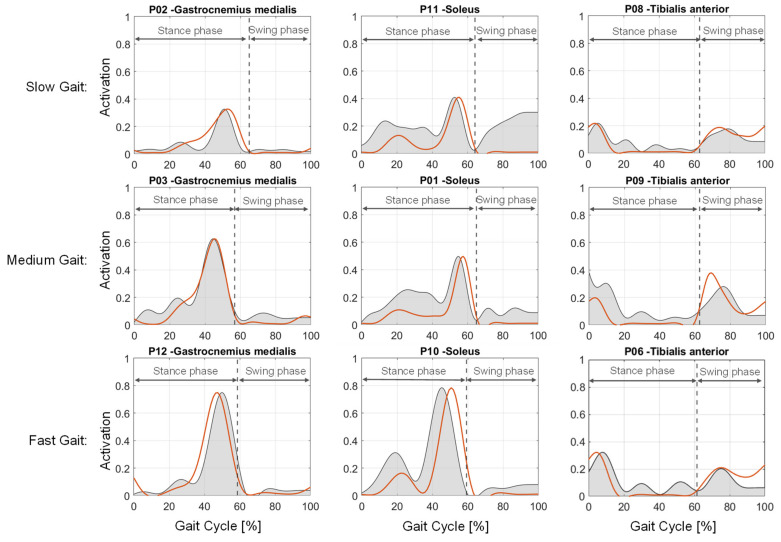
Comparison of simulated (orange) and measured EMG (grey) muscle activations of gastrocnemius medialis, soleus and tibialis anterior for different subjects at the three gait speeds: slow, medium and fast.

**Figure 8 ijerph-20-06687-f008:**
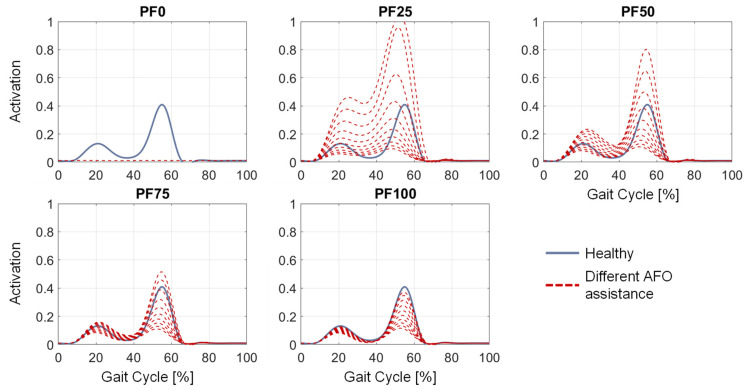
Resulting soleus muscle activations for P11 at slow gait speed with different AFO assistances (red) due to the parameter study in comparison to the healthy soleus muscle activation (blue).

**Figure 9 ijerph-20-06687-f009:**
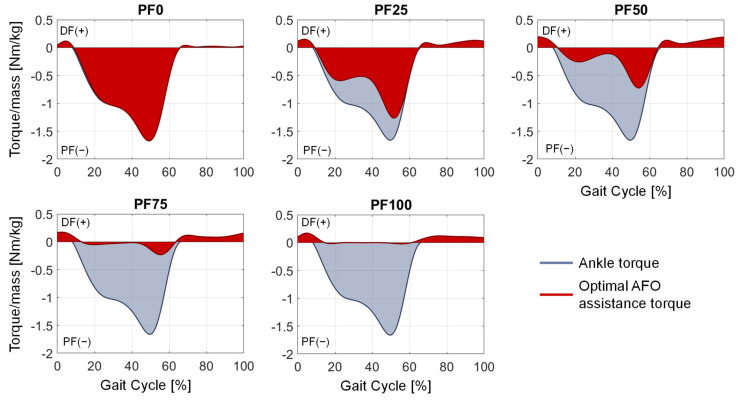
Optimal AFO assistance torque (red) compared to the acting ankle torque (blue). DF(+): dorsiflexion; PF(−): plantarflexion.

**Figure 10 ijerph-20-06687-f010:**
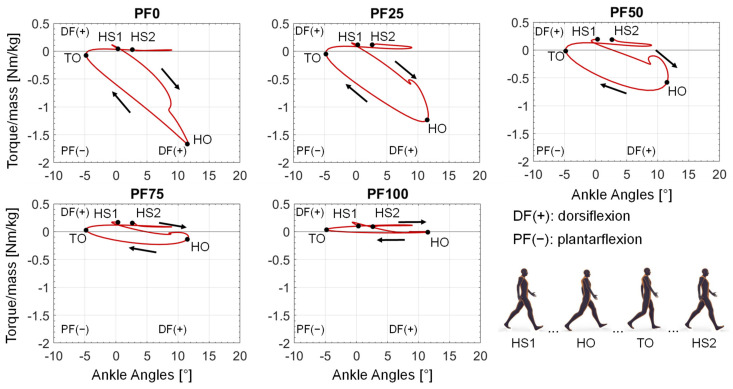
Resulting angle–torque curves of each optimal AFO assistance for each foot drop severity at slow gait speed. HS1: Heel Strike 1; HO: Heel-off; TO: Toe-off; HS2: Heel Strike 2.

**Figure 11 ijerph-20-06687-f011:**
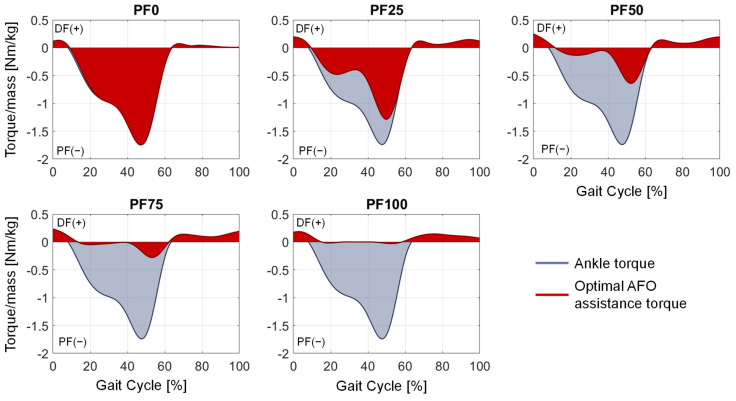
Optimal AFO assistance torque (red) compared to the acting ankle torque (blue) for medium gait speed.

**Figure 12 ijerph-20-06687-f012:**
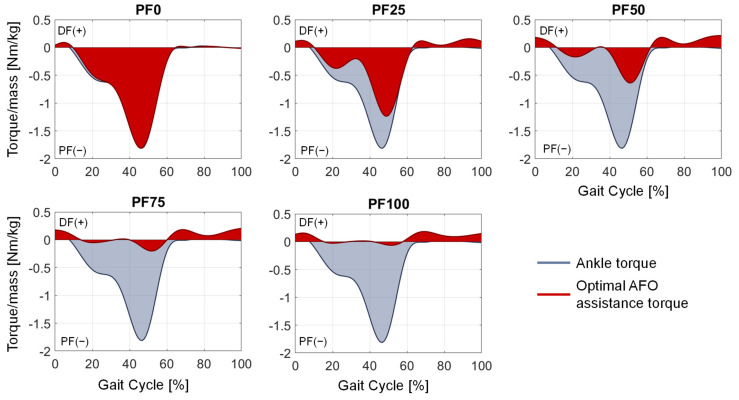
Optimal AFO assistance torque (red) compared to the acting ankle torque (blue) for fast gait speed.

**Figure 13 ijerph-20-06687-f013:**
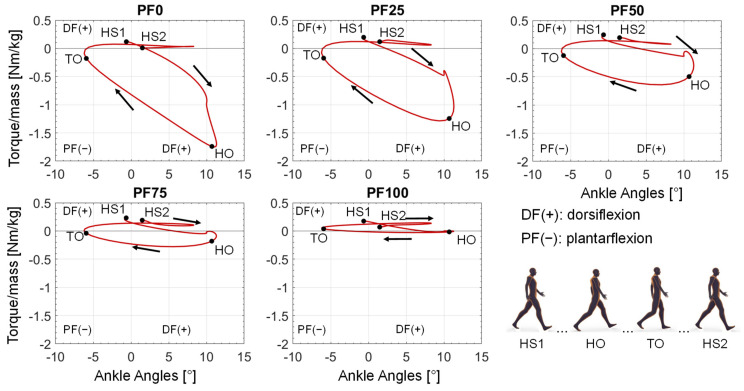
Resulting angle–torque curves of each optimal AFO assistance for each foot drop severity at medium gait speed. HS1: Heel Strike 1; HO: Heel-off; TO: Toe-off; HS2: Heel Strike 2.

**Figure 14 ijerph-20-06687-f014:**
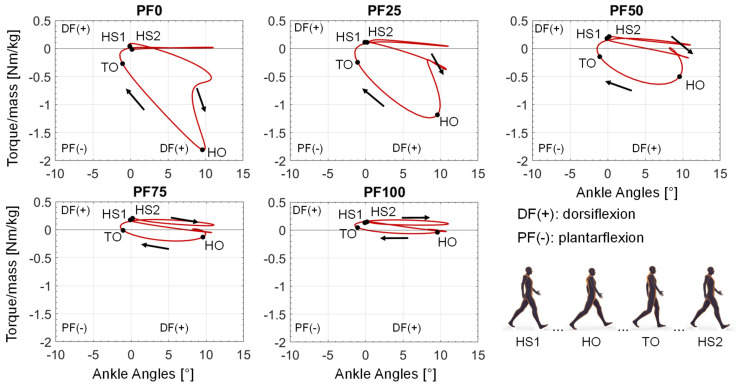
Resulting angle–torque curves of each optimal AFO assistance for each foot drop severity at fast gait speed. HS1: Heel Strike 1; HO: Heel-off; TO: Toe-off; HS2: Heel Strike 2.

**Table 1 ijerph-20-06687-t001:** Average percentage deviations of the plantarflexor muscles with the varied optimal forces of the ankle reserve actuator from the healthy plantarflexor muscle activations for each foot drop situation; green marks the smallest deviation for each foot drop severity, whereas red marks the largest one.

	Optimal Force—Ankle Reserve Actuator									
[%]	20	40	60	80	100	120	140	160	180	200
PF0	3.54	3.54	3.54	3.55	3.55	3.55	3.55	3.54	3.55	3.54
PF25	7.81	6.02	4.57	3.82	3.39	3.14	3.03	2.99	2.97	2.97
PF50	3.22	2.73	2.32	2.13	2.09	2.08	2.08	2.12	2.16	2.21
PF75	1.13	0.96	0.81	0.88	0.97	1.05	1.13	1.20	1.26	1.31
PF100	0.17	0.21	0.26	0.39	0.52	0.64	0.74	0.82	0.89	0.95

## Data Availability

Not applicable.

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
