# Peer review of "The Determination of Assistance-as-Needed Support by an Ankle–Foot Orthosis for Patients with Foot Drop"

_ijerph, 2023, doi:10.3390/ijerph20176687_

Round 1

Reviewer 1 Report

This is an article reporting a musculoskeletal model (MHM) to assist the manufacturing of customized wearable devices to “normalize” the gait of people with foot drop, resembling to a proof-of-concept analysis. The article is a little hard to read. Most importantly, I miss a subsection in the Methods section, called Analysis, to prepare me for interpreting the Results. Authors somewhat report part of their analysis in lines 259–263 but looks insufficient. Many reporting of the results is descriptive, but other involves comparing conditions (different gait speeds), percentages and deviations or agreements. How really different are the comparisons? Could it be only measurement error? How can we differentiate? Additionally, I’m struggling to understand the criteria used to evaluate the goodness of fit of the MHM. At the end of the article, I’m still trying to figure it out how good is this model to represent a potential solution to customize wearable devices to best assist the gait of patients with foot drop. 

SPECIFIC COMMENTS

Title

For a more informative Title, please add the study design to your title (PICOS approach)

Abstract

Line 7: “have issues with normal gait” is a lot vague. I suggest something like “impaired gait pattern functions”.

Line 9: Add “motor” before “requirements”.

Introduction

Line 29: revise this sentence. Peroneal nerve is not injured with central nervous system pathologies such as cerebral palsy or stroke you have stated above.

Line 50: “human’s healthy gait” sounds strange. Please, revise.

Methods

I feel that more information is necessary in the subsection “2.1.1 Experimental data” as if this experiment is going to be replicated. For example, calibration of the space for motion capture, which type and size of the electrodes, wireless or cable EMG (state the model), how was skin prepared to record EMG, etc. Summarizing, more clear data acquiring and processing reporting. 

Line 115: Report the height of the participants and BMI. Also, often these participants addressed as patients. Please revise so as not to confuse readers.

Line 141: replace “is” by “was”.

Make sure that all the constituents of the equations have been defined. Many have not.

Line 159: Replace “are” by “have been”.

Line 217: “in a gait equal to the gait of a healthy subject” is somewhat vague. “Do you mean the stride length of the steps, the movement pattern, the metabolic cost or other?

Lines 286–288: I should have known this before looking at the Results themselves, i.e., in a subsection of Methods, Analysis.

Lines 301–304: But how much do your data (dis)agree and disagree? For example, 2% of disagreement between two different scales for weighing 1 Kg of rice may be considered very good agreement. However, 2% difference for weighing gold or diamonds can be considered a lot of disagreement. Without solid agreement statistics, decision is (too) highly subjective, almost an opinion. Hence, what is the threshold value authors have set a priori to consider the one lowest average deviation value a good or poor agreement?

Line 330: Again, I should have known this before looking at the Results themselves.

Author Response

Dear Reviewer 1,

thank you very much for your detailed and comprehensive feedback to our manuscript, which heavily encouraged us to revise and improve our paper. You can find the revisions we have performed based on your comments in the attached file.

Yours sincerely,

The authors.

Reviewer 2 Report

The authors of this work demonstrated an approach for determining the required ideal support by an AFO during the gait cycle for a variety of patients, gait speeds and foot drop severities. With the help of musculoskeletal simulation, the assistance provided by the AFO affecting the resulting muscle activations in the lower leg muscles is investigated via a parameter study of the possible AFO strength. By identifying the assistance with the smallest deviation from the healthy muscle activation, the ideal support, referred to as the AAN support, can be established. Based on these results, concepts of an AFO that are capable of realizing the assistance of both rotational directions (dorsi- and  plantarflexion) in the ankle joint and accordingly provide the best possible treatment for foot drop patients can be further designed and also be tested for validation purposes.

The overall evaluation of he work is that it is well written and with comprehensive results.

Please consider the following minor comments.

1. Referencing must be consistent. In some places, the referencing were made with the author's name and reference number whereas most of the referencing were made only with reference number.

2. Need to write some texts between Section and its Subsection ( 2 and 2.1; 3 and 3.1)

Minor proof reading of the document is needed.

Author Response

Dear Reviewer 2,

thank you for the positive feedback of your review.

Encouraged by your comments, we have checked all references and citations. It was ensured that author names in combination with reference numbers are used, when a in-text-citation is used, for the regular citation only reference numbers are used as it is provided in the template file.

Furthermore, text was added between section 2 and 2.1 and also between section 3 and 3.1.

Yours sincerely,

The authors.

Round 2

Reviewer 1 Report

In general, I'm satisfied with the amendments authors have performed in the original manuscript. I believe the comprehensibility (and thereby the readability) of the new manuscript has been much improved. I have a few minor suggestions:

– replace "chapter" by "section" when you are referring to Introduction, Results, etc. (eg, line 107, line 292)

– Line 109: This is sentence is hard to read (though understandable). I suggest replacing "functionality" by "pattern" or "performance". 

– In Table 1, add to the title or create a footnote explaining what the greens and reds represents. I know that you have explained it in the text but the idea is that if readers only read the table, they can understand your findings as much as possible.

Best wishes.

Author Response

Dear Reviewer 1,

thanks for your positive feedback about the revised version. 

All of your suggested minor revisions have been incorporated into the new version.

Yours sincerely,

the authors.
